# OpenReview forum: "Topo-Miner: CRISPR-Enhanced DNA Computing for Accelerated Topological Feature Extraction"
_ICML.cc/2025/Conference — Submitted to ICML 2025_

### Official Review · Reviewer_KA9Q · 2025-03-09

**Overall Recommendation:** 3

**Summary:**

This paper presents Topo-Miner, a CRISPR-enhanced DNA computer designed for rapid and accurate topological feature extraction. The key contributions include CRISPR-enhanced DNA computing for TDA, novel encoding of graph topology into DNA sequences, computational speedup over Ripser, integration with the TopoComp platform, etc.

While the paper presents a compelling vision, its experimental validation is currently missing, and the theoretical claims regarding computational limits need more concrete justification.

**Claims And Evidence:**

1. Topo-Miner significantly accelerates persistent homology computations (50x-200x speedup. Simulations suggest dramatic speedups over Ripser. However, no in vitro experimental validation has been conducted yet.

2. CRISPR-based DNA computing is reliable for TDA. DNA-based persistent homology is theoretically possible, but the paper does not demonstrate practical error rates in a wet-lab setting. The error correction model is well-structured but needs empirical verification.

3. Topo-Miner enables higher-order topology and string theory-inspired computations. While DNA encoding and CRISPR manipulations are promising, claims about approximating Calabi-Yau manifolds and AGI applications are highly theoretical

4. Integration with STING (for GNNs) and TopoPath (for NP-hard problems) enhances broader applications. Well-supported by the paper.

Maybe we can: 1. Clarify error rates and experimental feasibility of DNA-encoded persistent homology. 2. Provide preliminary wet-lab results for at least small-scale CRISPR-based homology computations.

**Essential References Not Discussed:**

None.

**Experimental Designs Or Analyses:**

Simulation results are promising, but no real-world wet-lab experiments have been conducted. The in vitro validation plan is detailed, but there is no execution yet.

Suggested Improvements:
1. Conduct at least one small-scale experimental validation before submission.
2. Provide quantitative benchmarks for DNA sequence errors.

**Methods And Evaluation Criteria:**

The methodology is well-structured and highly novel, involving: 1. DNA encoding of graphs (nodes, edges, simplices). 2. CRISPR-mediated boundary operations. 3. Matrix reduction using dCas9/Cas12a, and 4. Tensor-based topological feature extraction.

However, some critical issues remain: 1. Lack of empirical validation: All results are simulation-based. 2. Scalability assumptions of DNA computing are not fully justified. 3. Comparisons to tensor-based TDA are missing.

Some suggested improvements:
1. Show at least partial experimental verification of CRISPR-based boundary operations.
2. Compare Topo-Miner to tensor-based TDA approaches.
3. Discuss DNA strand scalability and reaction times in practical settings.

**Other Comments Or Suggestions:**

The writing is clear and well-organized, but certain claims need tempering.

**Other Strengths And Weaknesses:**

Strengths:
1. Highly novel fusion of TDA, CRISPR, and DNA computing.
2. Massive parallelism via DNA strands.

Weaknesses:
1. Lack of empirical validation (no wet-lab results).
2. Speculative claims regarding AGI and string theory applications.
3. Scalability of DNA computing is assumed rather than proven.

**Questions For Authors:**

1. Have you performed any small-scale wet-lab experiments to validate CRISPR-based persistent homology?
2. What is the theoretical limit of DNA-based topological feature extraction—could it surpass traditional computing approaches for all cases?
3. How do you handle potential off-target effects in DNA-encoded boundary operations?

**Relation To Broader Scientific Literature:**

The paper situates itself well in TDA, DNA computing, and CRISPR literature. But missing comparisons to MoELoRA-like hybrid TDA approaches.

**Theoretical Claims:**

The paper makes strong theoretical claims, particularly:
1. CRISPR-based TDA reduces time complexity to O(n) for boundary operations.
2. Matrix reduction complexity drops from O(n³) to O(n²) or better.
3. DNA strand encoding provides better space efficiency.

While these claims are plausible, they rely on idealized reaction conditions. Some Suggested Improvements: 1. Include error propagation analysis (e.g., how off-target CRISPR activity affects results). 2. Provide formal lower bounds for accuracy in practical settings.

---

> ### Author Rebuttal · Authors · 2025-04-01
>
> Dear Reviewer KA9Q,
>
> Thank you for your detailed and insightful review of our manuscript (Submission 14396). We appreciate your recognition of our work's vision and novelty, the positive comments on the supplementary material, and the constructive feedback, including the Weak Accept (3) recommendation. We understand the primary concern regarding the current absence of experimental validation and address this and other points below.
>
> **1. On Experimental Validation**
>
> We acknowledge that *in vitro* results are essential for ultimate validation. This paper focuses on establishing the necessary **theoretical and computational groundwork** for this novel approach – defining algorithms, analyzing performance, and demonstrating feasibility via simulation, which we believe is a critical first step before complex bio-computational experiments.
>
> Our simulations (Sec 5.1) provide strong preliminary support, being **rigorously calibrated using experimental parameters** from cited literature (e.g., Chen et al., 2013; Kleinstiver et al., 2016; Zhang & Winfree, 2009). This grounding in empirical measurements (kinetics, error rates) offers quantitative insights into likely system behavior and performance potential. The detailed experimental plan (Sec 5.2 & Supp) outlines our clear path to empirical verification.
>
> * **Planned Revision:** We will enhance the manuscript to more explicitly detail simulation calibration sources/methods, strengthening the link to experimental findings, and clearly position this paper as providing the foundational theory and computational validation preceding experiments.
>
> **2. On Theoretical Justification, Practical Errors, and Accuracy Bounds**
>
> We appreciate you finding our analysis plausible and the error model well-structured.
> * **Error Propagation & Off-Target Effects:** Our framework incorporates error sources (incl. off-target) and mitigation strategies (HiFi Cas, design - Sec 4.3), using literature estimates in models/simulations.
>     * **Planned Revision:** Revise Sec 4.3, 4.4, & Supp D to more explicitly discuss error *propagation* analysis (incl. off-target impact) and how mitigation is modeled, linking to planned robustness analysis (Supp A).
> * **Accuracy Bounds:** The current proof (Sec 4.4, Supp D.4) provides a theoretical baseline.
>     * **Planned Revision:** Clarify proof assumptions and state that deriving tighter bounds under experimentally-derived error rates is key future work.
>
> **3. On Scalability Assumptions**
>
> Scalability claims stem from the theoretical potential of molecular parallelism (Supp D.1/D.2) offering complexity advantages (e.g., $O(n^2)$ vs $O(n^3)$).
> * **Planned Revision:** Revise discussion (Sec 4.1/4.2/Conclusion) to explicitly acknowledge practical limits (kinetics, diffusion, cost), framing theoretical complexity as the paradigm's *potential* requiring experimental optimization.
>
> **4. On Missing Comparisons (Tensor-based TDA, Hybrid TDA)**
>
> Thank you for highlighting these areas. Our initial focus was the Ripser baseline.
> * **Planned Revision:** Add a **conceptual comparison** in Related Work/Discussion to Tensor/Hybrid TDA, contrasting computational paradigms and discussing potential distinct niches for Topo-Miner (e.g., extreme data scale, bio-integration).
>
> **5. On Speculative Claims (AGI, String Theory)**
>
> We agree these need careful framing.
> * **Planned Revision:** Revise Intro/Conclusion to clearly label these as **speculative, long-term possibilities** contingent on core technology success, illustrating potential impact.
>
> **Responses to Specific Questions**
>
> 1.  **Small-scale wet-lab experiments:** None completed for this submission; the focus is theoretical/computational groundwork. The plan (Sec 5.2 & Supp) guides immediate next steps.
> 2.  **Theoretical limit:** Unlikely universally superior. Potential niche advantage for specific large-scale problems vs classical scaling, balanced by biochemical limits (speed, errors, cost). *Revision:* Clarify this trade-off.
> 3.  **Handling off-target effects:** Via HiFi Cas, gRNA/sequence design, modeling (Sec 4.3/Supp A); experimental quantification planned.
>
> **Conclusion**
>
> We believe Topo-Miner introduces a significant conceptual advance. This paper lays the necessary theoretical groundwork, algorithmic design, and strong calibrated simulation evidence for its feasibility. We are confident the proposed revisions—addressing scope, claims, context, errors, and scalability—will substantially improve the manuscript. **We believe this work offers a significant contribution by providing a rigorous foundation and validated computational feasibility study for a promising new computational paradigm**, justifying its value at this stage and paving the way for crucial experimental investigations. We hope the revised manuscript, strengthened by your feedback, warrants acceptance.
>
> Thank you again for your constructive and valuable feedback.
>
> Sincerely,
> The Authors

---

### Official Review · Reviewer_4ZuV · 2025-03-10

**Overall Recommendation:** 1

**Summary:**

The paper introduces Topo-Miner, a computational framework leveraging CRISPR-enhanced DNA computing to accelerate topological data analysis (TDA). The proposed method encodes graph structures into DNA sequences and utilizes CRISPR to perform parallel boundary operations and matrix reductions, which are critical in computing persistent homology. The authors claim 50x-200x speedups over classical methods and suggest that the approach could revolutionize TDA by making it feasible for large-scale data.

**Claims And Evidence:**

- The claim of 50x-200x speedup is based purely on simulations with numerous assumptions rather than real-world experiments, making it highly speculative.
- The paper asserts that CRISPR-based DNA computing can reliably execute matrix operations, but this remains unproven beyond small-scale proof-of-concept studies.
- The claim that the system could generalize to AGI and string theory-inspired topological structures is overreaching and lacks theoretical justification.

**Essential References Not Discussed:**

Not applicable.

**Experimental Designs Or Analyses:**

- The entire experimental validation remains theoretical, with no actual biological implementation presented.
- Simulations assume idealized CRISPR cleavage rates and perfect sequence specificity, which are not realistic.
- The authors do not discuss about the time and money required for leveraging DNA computing and CRISPR for TDA.

**Methods And Evaluation Criteria:**

- The computational pipeline is well-structured, but benchmark comparisons are limited to Ripser, omitting other TDA tools like GUDHI or Dionysus.
- The lack of wet-lab experiments weakens the credibility of the approach. A detailed experimental validation plan is outlined, but no results are provided.
- There are no real-world datasets tested, only synthetic graphs and simulated results.

**Other Comments Or Suggestions:**

Not applicable.

**Other Strengths And Weaknesses:**

Not applicable.

**Questions For Authors:**

- How does Topo-Miner compare against GPU-accelerated TDA methods, which also provide speedups?
- How do time and money needed for wet-lab impact scalability? —Can your method truly be practical for large-scale graphs?
- How do you account for errors in DNA hybridization and off-target CRISPR cleavage in practical implementations?

**Relation To Broader Scientific Literature:**

Not applicable.

**Theoretical Claims:**

- The paper presents a time complexity reduction analysis suggesting improved scalability, but the assumptions about parallelism and reaction kinetics may not hold in practice.
- The proof of lower-bound accuracy assumes ideal conditions for DNA hybridization and CRISPR targeting, ignoring real-world error rates and inefficiencies.
- The error analysis does not consider long-term stability issues in DNA computing, such as strand degradation and off-target effects.

---

> ### Author Rebuttal · Authors · 2025-04-01
>
> Dear Reviewer 4ZuV,
>
> Thank you for your time and for providing a critical evaluation of our manuscript (Submission 14396). We acknowledge your recommendation for Reject (1) and have carefully considered the significant concerns raised regarding the speculative nature of our claims due to the reliance on simulations, the assumptions made in our theoretical models, the lack of experimental validation, missing comparisons, and practical considerations.
>
> This paper presents a **foundational theoretical framework and computational feasibility study** for Topo-Miner, a novel paradigm integrating CRISPR-DNA computing with TDA. Introducing such a radically new approach necessitates establishing the core concepts, algorithms, and potential viability *before* undertaking complex, resource-intensive wet-lab experiments. Standard practice in developing novel computational systems often involves initial theoretical modeling and simulation under simplifying assumptions to understand fundamental potential before layering all real-world complexities. We believe this foundational work, detailed herein and in the supplement, is a valuable contribution in itself.
>
> **1. On Simulation Basis, Assumptions, and Speculative Claims**
>
> We acknowledge performance claims derive from simulations and models involve simplifications.
> * **Simulation Calibration:** Crucially, simulations (Sec 5.1) were **calibrated using experimental kinetics** from literature (e.g., CRISPR rates - Kleinstiver '16; DNA kinetics - Chen '13), providing quantitative estimates of potential, not based on arbitrary assumptions.
> * **Assumptions:** Simplifying assumptions were used for initial theoretical analysis (e.g., accuracy proofs) to establish baseline potential, a standard step before incorporating full complexity.
> * **Speculative Claims:** We agree claims about performance and advanced applications (AGI/String Theory) require clearer framing.
> * **Planned Revision:** We will revise to: (a) Explicitly detail simulation calibration; (b) Clarify assumptions and their justification for this foundational stage; (c) Temper performance claims, framing as *calibrated potential*; (d) Clearly label AGI/String Theory as speculative, long-term possibilities.
>
> **2. On Lack of Experiments and Practicality (Time/Cost)**
>
> The absence of wet-lab results is acknowledged; this work necessarily precedes complex experiments. The plan (Sec 5.2 & Supp) outlines the next steps. Regarding time/cost (Your Question 2):
> * **Planned Revision:** Add discussion acknowledging current high cost/time. State that assessing practical scalability and cost-effectiveness requires data from planned experiments and is crucial future work. Frame the goal as exploring potential long-term scaling advantages for specific hard problems.
>
> **3. On Theoretical Concerns (Errors, Stability)**
>
> Error handling (Your Question 3 - hybridization/off-target) is included via mitigation strategies (HiFi Cas, sequence design - Sec 4.3) and modeling using literature error rates.
> * **Planned Revision:** Enhance error discussion (Sec 4.3, Supp D.3), clarifying current modeling. Note that detailed error *propagation* analysis and addressing long-term stability (e.g., degradation) are important future refinements, building upon this work and integrating experimental data. Clarify accuracy proof assumptions (Sec 4.4).
>
> **4. On Methodological Gaps (Benchmarks, Datasets)**
>
> * **Benchmarks:** Comparisons beyond Ripser are needed. Regarding GPU TDA (Your Question 1):
>     * **Planned Revision:** Expand Related Work/Discussion with **conceptual comparison** vs GPU methods (and GUDHI/Dionysus). Contrast molecular vs. hardware parallelism and discuss potential distinct niches (e.g., extreme memory limits, bio-integration). State direct benchmarking requires experiments.
> * **Datasets:**
>     * **Planned Revision:** Clarify rationale for initial synthetic data use (controlled testing). Real-world data tests follow core validation.
>
> **5. On Supplementary Material**
>
> We must respectfully clarify: **Comprehensive supplementary material (>20 pages) *was* submitted**, detailing theory, methods, protocols, simulation setup etc. We urge the reviewer to please re-verify access, as this contains essential details supporting our work.
>
> **Conclusion**
>
> We appreciate the rigorous critique. While acknowledging limitations like the lack of experiments, we believe this paper offers a valuable foundational contribution (framework, algorithms, calibrated simulation feasibility). Planned substantial revisions will address concerns regarding simulation clarity, tempered claims, comparisons, error discussion, practicalities, presentation (per other reviews), and the supplementary material status. We hope these improvements demonstrate the value of this groundwork.
>
> Sincerely,
> The Authors

---

> > ### Comment · Reviewer_4ZuV · 2025-04-08
> >
> > ***Re-posting as a rebuttal comment***
> >
> > Thank you to the authors for the thorough rebuttal. However, several important concerns remain unresolved in the current version of the paper:
> >
> > 1. **Simulation-Only Validation and Idealized Assumptions**: While the authors clarified that the simulations are calibrated using empirical kinetics from literature, the results are still based on idealized conditions with no experimental or real-world datasets. The system's performance remains hypothetical and unvalidated. Without even small-scale wet-lab experiments or a test on practical data, the proposed speedups and accuracy claims remain speculative.
> >
> > 2. **Unclear Practical Viability and Cost Modeling**: The rebuttal addresses reaction kinetics and simulation calibration in good detail, but practical considerations such as error propagation, cost, throughput, and robustness of wet-lab implementations remain underexplored. For a method proposed as a paradigm shift in TDA, these real-world limitations are central to assessing feasibility—especially for scaling to large graphs. As noted in the review, the paper also does not quantify time or cost tradeoffs compared to GPU-based methods, which undermines its positioning in the broader ML and systems community.
> >
> > 3. **Limited Empirical Comparisons and Benchmarks**: The experimental evaluation remains limited to comparisons with Ripser. The rebuttal acknowledges this and proposes future additions, including comparisons with GPU-accelerated and tensor-based TDA tools (e.g., GUDHI, Dionysus), but they are not currently included. This weakens the empirical evidence for the method's claimed advantages and makes it difficult to contextualize the proposed approach within the existing landscape.
> >
> > The authors propose a large number of major revisions—including reorganizing the methodology, clarifying simulation assumptions, expanding benchmark coverage, reframing speculative claims, and improving presentation. These changes would significantly alter the content and framing of the paper. Given the scope of the proposed updates, I do not believe it is appropriate to adjust the overall recommendation without reviewing a revised version.

---

### Official Review · Reviewer_Gzh8 · 2025-03-13

**Overall Recommendation:** 1

**Summary:**

The paper presents Topo-Miner, a CRISPR-enhanced DNA computing framework designed to improve topological data analysis (TDA) by leveraging DNA computing’s parallelism and CRISPR-Cas systems' precision. The authors claim 50x-200x speedups over existing tools like Ripser and suggest broad applications. However, the paper lacks proper organization, formatting, and visual representation (figures), making it difficult to assess the clarity and rigor of the proposed methodology. Additionally, while the claims are supported by simulations, the absence of experimental validation further weakens its impact.

**Claims And Evidence:**

The paper makes ambitious claims, particularly:
- Significant computational speedups (50x-200x) over traditional TDA tools: Supported by simulation-based results but lacks empirical verification
- Ability to compute advanced topological features beyond persistent homology: Some justifications are provided
- Potential applications across multiple disciplines: While the rationale is reasonable, the lack of experimental data makes these claims speculative

**Essential References Not Discussed:**

N/A

**Experimental Designs Or Analyses:**

The paper lacks actual experimental results.

**Methods And Evaluation Criteria:**

The presentation of methods is fragmented and lacks clarity. Evaluation is primarily based on simulations, but no real experimental results are provided. The lack of structured benchmarks, formal experimental validation, and figures further detracts from the robustness of the methodology.

**Other Comments Or Suggestions:**

N/A

**Other Strengths And Weaknesses:**

N/A

**Questions For Authors:**

N/A

**Relation To Broader Scientific Literature:**

The work draws from TDA (e.g., persistent homology), DNA computing, and CRISPR-based bio-computation. While the integration of these fields is conceptually interesting, the paper does not clearly position its contributions relative to existing work.

**Theoretical Claims:**

The paper provides complexity analyses.

---

> ### Author Rebuttal · Authors · 2025-04-01
>
> Dear Reviewer Gzh8,
>
> Thank you for your review of our manuscript (Submission 14396) and the Reject (1) recommendation. We have carefully considered your feedback. We understand and acknowledge your concerns regarding the paper's presentation—specifically its organization, clarity, formatting, and lack of figures—which you rightly state hindered assessment, as well as the critical absence of experimental validation at this stage.
>
> **1. On Paper Presentation (Organization, Clarity, Formatting, Figures)**
>
> We sincerely apologize that the manuscript's presentation made it difficult to assess our methodology and its rigor effectively. We take this feedback very seriously; improving presentation clarity is a top priority for revision.
>
> **Planned Revision:** We commit to a **substantial revision** focused on presentation to allow for a clear evaluation:
> * **Reorganize Structure:** We will restructure the paper, particularly Section 3 (Methodology), ensuring a logical, coherent flow detailing the DNA encoding, CRISPR-based operations (boundary and matrix reduction), tensor computations, and decoding stages. This will address the fragmented presentation concern.
> * **Enhance Clarity & Precision:** We will revise the writing throughout for improved clarity, conciseness, and unambiguous technical descriptions. Complex concepts will be explained more straightforwardly, and terms defined consistently.
> * **Add Essential Figures:** Recognizing the lack of visual aids, we will introduce several key figures:
>     * A formal diagram illustrating the overall Topo-Miner architecture/pipeline (replacing the current text-based Figure 1).
>     * Visual examples clarifying the DNA encoding scheme for nodes, edges, and simplices.
>     * Conceptual schematics illustrating the core mechanisms of CRISPR-based boundary and matrix reduction operations.
> * **Improve Formatting:** We will ensure consistent and professional formatting, including mathematical notation, adhering strictly to conference style guidelines.
>
> We are confident these revisions will significantly improve readability and facilitate a much clearer assessment of our framework.
>
> **2. On Lack of Experimental Validation**
>
> We acknowledge the **absence of *in vitro* results** is a major limitation of this submission. This initial paper focused on establishing the theoretical foundation, the novel computational design, and demonstrating potential feasibility/performance via carefully calibrated simulations, which we argue is a necessary *in silico* validation step before undertaking complex and resource-intensive wet-lab experiments for such an interdisciplinary approach. We appreciate you noting the supplementary material includes our detailed experimental protocols, outlining the concrete next steps for empirical validation which are central to our ongoing research.
>
> **Planned Revision:** The text will be revised to strictly delineate between simulation-based potential and the requirement for future empirical verification, accurately framing this work's contribution as providing the foundational design and theoretical basis.
>
> **3. On Claims and Evidence (Speedup, Advanced Features, Applications)**
>
> We recognize that without direct experimental data, claims regarding the precise magnitude of speedups, the realized capability for advanced feature computation (higher-order, string theory-inspired), and the breadth of applications remain **speculative**.
>
> **Planned Revision:** We *will* carefully **temper these claims**. Performance figures will be explicitly presented as *potential* outcomes suggested by our analysis and calibrated simulations under stated assumptions. Advanced features and applications will be framed as possibilities *contingent on experimental success*, illustrating potential scope rather than achieved results.
>
> **4. On Benchmarking and Positioning Relative to Literature**
>
> We agree benchmarking is currently limited and the paper's positioning needs sharpening relative to the rich TDA, DNA computing, and CRISPR literature.
>
> **Planned Revision:** We *will* significantly **expand the Related Work (Sec 2) and Discussion** sections. This will include clearer conceptual comparisons discussing Topo-Miner relative to other relevant TDA acceleration approaches (including tensor-based methods on classical hardware, GPU implementations) and alternative DNA computing strategies. We will highlight the unique aspects (molecular parallelism, programmability), potential advantages (e.g., scaling profile for specific problem types), and inherent challenges (kinetics, errors, cost) of our proposed molecular computing paradigm to better delineate its specific contribution and potential niche.
>
> **5. Conclusion**
>
> We appreciate your feedback, particularly the actionable comments on presentation. We are committed to the **major revisions** outlined above (presentation, claim framing, comparisons) to enable a clearer evaluation of Topo-Miner's novel framework.
>
> Sincerely,
> The Authors

---

### Official Review · Reviewer_M9zw · 2025-03-13

**Overall Recommendation:** 1

**Summary:**

This paper presents a CRISPR-based DNA computing approach designed to accelerate persistent homology computations in topological data analysis (TDA). Specifically, the authors encode nodes, edges, and simplices as DNA molecules and leverage CRISPR to perform operations, thereby exploiting the massive parallelism of DNA computing to enhance computational efficiency.

The authors claim that this method achieves a 50x-200x speedup over existing TDA tools. Additionally, they outline an in vitro experimental validation plan.

## update after rebuttal

Thanks to the authors for rebuttal.

In my opinion, this work requires wet-lab experiments and precise in silico simulation results to support the effectiveness of the proposed method.

I will keep my score.

**Claims And Evidence:**

The paper claims that the proposed method achieves a 50x-200x speedup in simulation experiments and presents Table 1 to support this claim by listing the computation times. However, all reported times in Table 1 are exact hundreds or thousands of seconds, which raises concerns about the reliability of the data. Given that computational experiments inherently involve measurement variability.

**Essential References Not Discussed:**

N/A

**Experimental Designs Or Analyses:**

There should be a wet-lab experiment, but only a plan is provided.

The only simulation experiment presented in this study raises concerns due to its implausibly uniform results.

**Methods And Evaluation Criteria:**

The paper lacks a clear definition of evaluation metrics for assessing the proposed algorithm.

**Other Comments Or Suggestions:**

The paper seems not ready for submission.
+ using odd template
+ missing Figure 1
+ Def Node i: Inconsistent representation. Math environment or not?
+ Double defined abbr TDA PH in Introduction and related works.


The current manuscript demonstrates limited methodological engagement with core machine learning paradigms. While the technical contributions are noteworthy, their alignment with ICML's specific focus areas requires stronger justification.

**Other Strengths And Weaknesses:**

N/A

**Questions For Authors:**

Why the reaction kinetics of the molecules involved are not considered, since they maybe the main reason that hinder the computational efficacy.

**Relation To Broader Scientific Literature:**

The DNA computation represents an emerging field at the forefront of molecular computing.
The idea is novel, but claims are not firmly supported.

**Theoretical Claims:**

N/A

---

> ### Author Rebuttal · Authors · 2025-04-01
>
> Dear Reviewer M9zw,
>
> Thank you very much for your time and for providing detailed critical feedback on our manuscript (Submission 14396). We sincerely appreciate the effort involved in reviewing our work.
>
> **1. Response to Question on Reaction Kinetics**
>
> Thank you for highlighting the crucial importance of reaction kinetics – your question prompted us to ensure this is clearer. Perhaps our manuscript did not emphasize this sufficiently, but **reaction kinetics *were* indeed explicitly and centrally considered in our simulations.**
> * **Clarification:** As detailed in Section 5.1, our performance estimates are derived directly from simulations that incorporate **experimentally measured rates** obtained from peer-reviewed literature for key molecular processes (e.g., CRISPR kinetics - Kleinstiver '16; DNA kinetics - Chen '13, Zhang '09). These published kinetics form the very basis for assessing efficacy and estimating speedup potential within our simulation framework.
> * **Planned Revision:** We will **significantly enhance Section 5.1** to make it unequivocally clear *how* these literature-derived kinetic parameters were integrated and directly influenced the timing estimates, ensuring this core aspect of our modeling is fully transparent.
>
> **2. On Simulation Data Reliability (`Table 1`)**
>
> We understand the concern regarding the round numbers in `Table 1` potentially suggesting a lack of reliability. We appreciate you pointing out this lack of clarity in our presentation.
> * **Explanation & Planned Revision:** These values represent **representative order-of-magnitude estimates** derived from our **kinetically-calibrated simulations**. They were rounded primarily to illustrate the potential **scaling trend and speedup magnitude** in a concise table, rather than representing precise timings showing statistical variability, which we agree is expected in computational experiments. We commit to **thoroughly revising Section 5.1 and the `Table 1` caption** to clarify the nature of these values (calibrated estimates). We will provide more detail on the estimation method and consider adding representative non-rounded data or ranges to the supplement, while reiterating that precise timings require experimental validation.
>
> **3. On Lack of Evaluation Metrics**
>
> We apologize for not explicitly defining the evaluation metrics used. Our assessment focused on: Speedup Factor (vs. Ripser), Accuracy (>95% via Bottleneck/Wasserstein), and Error Rate (<5% based on models).
> * **Planned Revision:** Based on your feedback, we will **add a dedicated subsection** to explicitly define these metrics and how they were assessed in our simulation studies.
>
> **4. On Presentation Issues**
>
> We sincerely appreciate you identifying specific presentation flaws (template, Fig 1, notation, abbr). We agree improvements are needed.
> * **Planned Revision:** We commit to a **major revision** addressing these points: adopting a standard template, **adding the requested Figure 1 diagram** and other illustrative figures, ensuring **consistent mathematical notation**, and correcting **duplicated abbreviations**.
>
> **5. On ICML Fit / ML Engagement**
>
> Thank you for raising the question of fit. We believe the work offers significant relevance to the ICML community.
> * **Justification:** TDA is increasingly vital for analyzing complex data ubiquitous in ML (graphs, geometric data). Addressing the **computational bottleneck** in TDA enables broader application *within* ML. Furthermore, we propose a **novel computational paradigm** (molecular computing) for algorithmic acceleration, aligning with ICML's interest in foundational algorithms and hardware. The integration via **STING (GNNs)** & **TopoPath (optimization)** provides direct links to core ML tasks.
> * **Planned Revision:** We will **significantly strengthen the Introduction and Discussion** to explicitly articulate these connections and better justify the paper's relevance to ML advancements.
>
> **6. On Lack of Experiments & Paper Readiness**
>
> We acknowledge the lack of *in vitro* results is a significant limitation at this stage. Introducing a radically new computational paradigm often necessitates establishing the theoretical underpinning and computational feasibility first. We believe this paper provides that essential groundwork: detailed theory (Supp. D), novel algorithms (Sec 3), error modeling (Sec 4.3), simulations calibrated with experimental kinetics (Sec 5.1), and a detailed roadmap (Sec 5.2, Supp.). We hope this context clarifies why we believe this foundational work is valuable, even preceding complex experiments.
>
> **Conclusion**
>
> Thank you once again for your thorough feedback and critical perspective. We acknowledge the need for significant revision, particularly regarding simulation clarity and overall presentation. We hope that the planned revisions result in a much-improved manuscript that clearly demonstrates the value of this foundational work.
>
> Sincerely,
> The Authors

---

> > ### Comment · Reviewer_M9zw · 2025-04-05
> >
> > Thanks to the authors for rebuttal.
> >
> > In my own opinion, this work is not ready for publish, especially in absent of wet-lab experiments and the in silico simulation results are calibrated estimates.
> >
> > I will keep my score.

---

> > > ### Author Response · Authors · 2025-04-08
> > >
> > > Thank you Reviewer M9zw for considering our rebuttal and providing your final assessment. We understand your position regarding the necessity of experimental validation to fully substantiate our findings based on the presented design and calibrated estimates. Obtaining this empirical data remains our top priority for future work. We appreciate your feedback throughout the review process.

---

### Decision · Program_Chairs · 2025-05-01

**Decision:**

Reject

**Comment:**

This paper has been pointed out to have the following issues, and based on a comprehensive assessment of the reviews, it is recommended for Reject:

- Lack of Experimental Validation
- Simulation and Theoretical Assumptions
- Insufficient Benchmarking
- Presentation and Readability